# CAMA: A New Framework for Safe Multi-Agent Reinforcement Learning Using Constraint Augmentation

## Abstract

With the widespread application of multi-agent reinforcement learning (MARL) in real-life settings, the ability to meet safety constraints has become an urgent problem to solve. For example, it is necessary to avoid collisions to reach a common goal in controlling multiple drones. We address this problem by introducing the Constraint Augmented Multi-Agent framework — CAMA. CAMA can serve as a plug-and-play module to the popular MARL algorithms, including centralized training, decentralized execution and independent learning frameworks. In our approach, we represent the safety constraint as the sum of discounted safety costs bounded by the predefined value, which we call the safety budget. Experiments demonstrate that CAMA can converge quickly to a high degree of constraint satisfaction and surpasses other state-of-the-art safety counterpart algorithms in both cooperative and competitive settings.

## 1 Introduction

Multi-agent problems are ubiquitous in real world, such as robotics (Al-Abbasi et al., 2019; Mguni et al., 2021), transportation systems (Zhou et al., 2020; Chu et al., 2019), network optimization (Wang et al., 2020; Wai et al., 2018), and multi-player video games (Du et al., 2019; Samvelyan et al., 2019; Han et al., 2019; Peng et al., 2017). A modern approach to solving these decision-making problems is multi-agent reinforcement learning (MARL), which tackles these problems using only interactions with the environment. There are many different frameworks within MARL such as fully centralized Berner et al. (2019); Sukhbaatar & Fergus (2016), independent learning (IL) de Witt et al. (2020); Zhang et al. (2018) and a hybrid framework which is the centralized training and decentralized execution (CTDE) (Foerster et al., 2018; Lowe et al., 2017; Yang et al., 2018). However, within the deployment of MARL, safety is still a crucial problem, which has not been fully solved yet.

In recent years, several works have incorporated safety constraints into RL training, such as optimizing policy under constraints (Di Castro et al., 2012; Tessler et al., 2018; Achiam et al., 2017; Chow et al., 2018), adding safety layers (Dalal et al., 2018), or constructing verifiable safe exploration (Anderson et al., 2020), etc. In the context of safe MARL, recent papers extend constrained policy optimization (Achiam et al., 2017) to multi-agent domain (Gu et al., 2021) as a model-free safe MARL algorithms. But there are still challenges with low reward performance compare to the non-safe MARL algorithms. There are also some works performed constrained policy optimization by transforming it into a min-max game (Lu et al., 2021; Liu et al., 2021). However, which limited by the specific framework, it cannot generalize to other framework such as solving the competitive game. Therefore, a more general safe MARL framework with high reward performance is still lacking at this stage.

To fill this gap, in this paper we propose a general module that can be incorporated into different MARL algorithms. The proposed **C**onstraint **A**ugmented **M**ulti-**A**gent framework, coined as **CAMA**, is a plug-and-play method to improve cutting-edge non-safe MARL algorithms satisfying the adding constraints. Furthermore, CAMA aims to address both cooperation and competitive setting under CTDE and IL frameworks. In our algorithm, we represent the safety constraint as the sum of discounted safety costs bounded by a pre-defined scalar, which we call the safety budget.

The main idea of this approach is the introduction of the hazard value, which tracks the accumulated costs and represents the remaining safety budget. When the hazard value falls below zero, CAMA assigns a low negative reward to the agents, incentivizing them to learn a safe policy. The implementation of CAMA can be seen as a direct modification of the environment and indirectly influencing the algorithm by constraints augmentation so that there is no need for any new algorithm-based assumptions.

In summary, our contributions are three-fold. Firstly, CAMA is a flexible framework with a plug-and-play feature, which can be combined with many existing MARL algorithms. Secondly, CAMA can work in both CTDE and IL settings, including cooperative and competitive multi-agent games. Lastly, we evaluate CAMA on a series of multi-agent control tasks in SMAMujoco (Gu et al., 2021) and Gym Compete (Bansal et al., 2017). Empirical results demonstrate the effectiveness of our solutions both in terms of constraint satisfaction and reward maximisation compared to their state-of-the-art counterparts.

## 2 RELATED WORK.

In the following, we review the related works on general MARL, MARL with constraint and state augmentation method.

Existing MARL algorithms are often developed under the paradigm of CTDE and IL (Oroo-jlooyJadid & Hajinezhad, 2021): CTDE is a commonly used learning framework, which updates decentralized policies by using the centralized critic architecture (Du et al., 2019; Gu et al., 2021). In CTDE, the joint critic network is based on the all agents' states and actions, and thus generally handles the joint team reward scenarios (Kuba et al., 2022; Yu et al., 2021). Conversely, if the setting is a competitive game or only focus on the individual reward, then IL is another MARL paradigm to be based on. The earliest discussions of IL in a multi-agent-based environment can be traced back to (Tan, 1993). It subsequently evolved into the IL algorithm using neural networks as function approximators (Foerster et al., 2018; Rashid et al., 2020). Some recent attempts, like de Witt et al. (2020) extend single agent proximal policy optimization algorithms (Schulman et al., 2017) into the multi-agent IL setting.

Although MARL has received significant attention in recent years, there are still many unresolved safety related challenges (Gu et al., 2022), such as the multiple constraints setting, low algorithm efficiency problem etc. There are generally several ways to solve additional safe constraints. Recent attempts such as MACPO, MAPPO-L(Gu et al., 2021) proposed to fill such a gap as the first safe model-free MARL algorithms, which are extensions of CPO (Achiam et al., 2017) and HATRPO (Kuba et al., 2022), respectively. However, neither MACPO nor MAPPO-L is guaranteed to be applied in competitive games, which resulting in limited scalability. Another research direction is based on the parameters-sharing hypothesis. For example, the CMIX (Liu et al., 2021) can combine multi-objective programming and the Q-mix framework to solve the constraint MARL problem. However, in CMIX, different Q-function approximators are required for each constraint and each agent, which leads to scalability and efficiency challenges. Another parameters-sharing based approach, Safe Dec-PG (Lu et al., 2021) aims to stratify the constraint by passing the parameters through a predefined communication network. Instead, our framework requires no communication during policy execution. Another route to avoid unsafe action is to use shielding and barrier functions, such as ElSayed-Aly et al. (2021); Cai et al. (2021). However, those approaches require pre-training or strong prior knowledge to create the shields for filtering actions. Moreover, they cannot generalize to new scenarios where safety shield is not known. In contrast, CAMA is more flexible when dealing with new constrained environments. We test different types of tasks with varying agents without designing or pre-train a particular shielding function for each task.

The state augmentation method extend the specific state to an environment, in order to enhance policy performance or satisfy certain constraints (Calvo-Fullana et al., 2021). Recent works like (Qiu et al., 2021) augmented the CVaR to measure over the learned distributions of individuals' Q values, Chen et al. (2020) and Foerster et al. (2017) augmented delay awareness and experience replay, respectively. The idea of enhancing safety-related variables has been considered in the past, e.g., in classical control methods (Daryin & Kurzhanski, 2005) and in single-agent safe RL (Sootla et al., 2022a;b; Chow et al., 2017). We apply it in a multi-agent framework, but multiple constraints and multiple policy settings hinder the direct extension. Some works, such as (Chen et al., 2020;

Foerster et al., 2017), also apply augmentation methods to MARL problems, but we are the first to augment the constraint to solve the safe MARL problem.

## 3 PRELIMINARIES

**Constrained Multi-agent MDP (CMMDP)** A CMMDP (Boutilier, 1996) extends CMDP (Ma et al., 1986) by adding a factored action space, which can be defined by a tuple $\mathcal{M} = \left\langle \mathcal{N}, \mathcal{S}, \mathcal{A}, \mathcal{P}, \gamma_r, \gamma_c, \rho^0, \mathcal{R}, \mathcal{C}, \mathcal{B} \right\rangle$. Here, $\mathcal{N} = \{1, \ldots, n\}$ is the set of agents, $\mathcal{S}$ denotes the state space, and $\mathcal{A} = \prod_{i=1}^{n} A^i$ the joint action space, $\mathcal{P}(s' \mid s, a) : \mathcal{S} \times \mathcal{A} \times \mathcal{S} \to [0, 1]$ is the probabilistic transition function. $\gamma_r, \gamma_c \in (0, 1)$ are the discount factor for reward and cost. $\mathcal{R}(s, \boldsymbol{a}) : \mathcal{S} \times \mathcal{A} \to \mathbb{R}$ is the reward function of state $s$ and joint action $\mathbf{a}$, $\mathcal{C} = \left\{ C_j^i \right\}_{1 \le j \le m^i}^{i \in n}$ is set of safety costs, where $C_j^i$ is the $j^{th}$ element of the total $m^i$ constraints, $C_j^i : \mathcal{S} \times \mathcal{A}^i \to \mathbb{R}$. $\mathcal{B} = \left\{ b_j^i \right\}_{1 \le j \le m^i}^{i \in \mathcal{N}}$ is the set of corresponding safety budgets, the maximum value of agents can be violated, $\rho^0$ is the initial state distribution. Denote agent $i$'s policy as $\pi_i : \mathcal{S} \times \mathcal{A}_i \to [0, 1]$, $\boldsymbol{\pi} = \{\pi_i\}_{i=1}^n$. CMMDP aims to find a optimal joint policy $\boldsymbol{\pi}^*$ that maximizes the joint reward $J_{\mathcal{R}}(\pi) \triangleq \mathbb{E}_{s_0, \boldsymbol{a}_0, \ldots} \left[ \sum_{t=0}^{\infty} \gamma_r^t \mathcal{R}(\boldsymbol{s}_t, \boldsymbol{a}_t) \right]$ while satisfying the safety constraints $J_{\mathcal{L}}(\pi, i, j) \triangleq \mathbb{E} \left[ \sum_{t=0}^{\infty} \gamma_c^t C^i \left( \boldsymbol{s}_t, \boldsymbol{a}_t^i \right) \right] \le b^i$, where $C^i = \sum_j C_j^i$, $s_0 \sim \rho^0, \boldsymbol{a}_t^i \sim \pi_i(\cdot \mid \boldsymbol{s}_t), \boldsymbol{s}_{t+1} \sim \mathcal{P}(\cdot \mid s_t, \boldsymbol{a}_t)$ for all $i \in \mathcal{N}$.

Two commonly used architectures are **Centralized Training with Decentralized Execution** (**CTDE**) and **Independent Learning** (**IL**). CTDE update independent policies by maximizing the expected team reward $J_{\mathcal{R}}$ using the policy gradient. IL decomposes an $n$-agent MARL problem into $n$ decentralized single-agent problems, and ignores other agents by treating them as part of the environment. The policy update is conditioned on the local observations or global state and actions $\pi_i : s_i \times \boldsymbol{a}_i \to \mathbb{R}$, then received the local reward and cost. CTDE is widely used in cooperative tasks with a shared team reward, and IL can work in both cooperative and competitive settings.

**Constraints State Augmentation.** State augmentation eliminates safety constraints by incorporating them into the state space (Chow et al., 2017; Sootla et al., 2022a). Defining $z_t$ as a scaled version of the remaining safety budget $b$, where $z_t$ can be calculate as $\boldsymbol{z}_{t+1} = (\boldsymbol{z}_t - c(\boldsymbol{s}_t, \boldsymbol{a}_t))/\gamma_c, \boldsymbol{z}_0 = b$. Using this new variable, they reshape the reward by

$$\widetilde{R}(s_t, \boldsymbol{z}_t, a_t) = \begin{cases} R(s_t, a_t) & \boldsymbol{z}_t \ge 0 \\ k & \boldsymbol{z}_t < 0 \end{cases} \tag{1}$$

where $k<0$ is an unsafe reward. Using these simple modifications, Sootla et al. (2022a) showed that an algorithm solving this problem actually solves a safe RL with probability one constraints under certain conditions.

## 4 METHODOLOGY

In this section, we first define the basic definition of CAMA-MDP and list the basic assumptions and propositions required. After that, we introduce the specific derivation process of CAMA under the CTDE and IL paradigms in subsections 4.2 and 4.3. Finally, we made an algorithm to summarize the workflow of CAMA.

### 4.1 THE DEFINITION OF CAMA

We formalize our approach by considering the following definition,

**Definition 1** . Given a CMMDP, we define a Constraint Augmented Multi-Agent Markov Decision Process (**CAMA**-MDP) as a tuple: $\widetilde{\mathcal{M}} = \left\langle \mathcal{N}, \widetilde{\mathcal{S}}, \mathcal{A}, \widetilde{\mathcal{P}}, \gamma_r, \gamma_c, \widetilde{\mathcal{R}}, \mathcal{C}, \mathcal{B} \right\rangle$, where $\widetilde{\mathcal{S}} = \mathcal{S} \times \boldsymbol{h}_t$ is the constraint augmented state space, $\boldsymbol{h}_t = \{h_t^i\}_{i=1}^n$ denotes the set of hazard value, $\widetilde{\mathcal{P}} : \widetilde{\mathcal{S}} \times \mathcal{A} \times \widetilde{\mathcal{S}} \to \mathbb{R}$. Let $\widetilde{\mathcal{R}} = \frac{1}{n} \sum_{i=1}^n \widetilde{\mathcal{R}}_i$ denotes reward function, where $\widetilde{\mathcal{R}}_i = \widetilde{\mathcal{S}} \times \mathcal{A} \to [0, +\infty)$ is the $i^{th}$ element of $\mathcal{R}$ defined in the tuple.

To guarantee the optimal policy of CAMA-MDP exists and is also the same as CMMDP's $\pi^*$, we can adapt the theoretical results: Assumption.A1-A3 from Sootla et al. (2022a) in a straightforward

manner for CAMA IL frameworks. But in the CTDE setting, we still need the following regularity assumption, in particular, to guarantee that the policy set is closed and bounded:

**Assumption 1** *There exists $\theta \in \mathbb{R}$, such that $0 < \theta \ll 1$, and for every agent $i \in \mathcal{N}$, the policy space $\Pi^i$ is $\theta$-soft, which means for every $\pi^i \in \Pi^i$, $s \in \mathcal{S}$, and $a^i \in \mathcal{A}^i$, we have $\pi^i\left(a^i \mid s\right) \geq \theta$.*

This assumption has been discussed in Lemma 3 (Kuba et al., 2022). For more details, we provided the theoretical analysis in Appendix A.

## 4.2 CAMA IN CTDE

Let us firstly consider a CMMDP problem with multi-constraints and joint reward. Since the $J_{\mathcal{C}}(\boldsymbol{\pi}, i, j)$ always greater than 0, which can be equivalent to enforcing the finite number of the following constrains:

$$\sum_{k=0}^{t} \gamma_c^k C_j^i\left(s_k, \boldsymbol{a}_k\right) \leq b_j^i, \forall t \geq 0, \tag{2}$$

Therefore, when an agent violates the limit at a specific moment $t_v$, it will be in a hazard zone for a series of subsequent time points $t \geq t_v$, which allows us to incorporate constraints into the instantaneous time as the safe cost, taking safety into account when solving the task. In the multi-agent case, defining *the hazard value* is slightly more complicated due to the number of constraints. We introduce a hazard value for each constraint and each agent as follows,

$$\boldsymbol{h}_{j,t}^i = \left(b_j^i - \sum_{k=0}^{t} \gamma_c^k C_j^i\left(s_k, a_k^i\right)\right) / \gamma_c^t. \tag{3}$$

For simplicity, we will use the notation $\boldsymbol{h}_t^i$ for the vector $\left(\boldsymbol{h}_{0,t}^i \quad \cdots \quad \boldsymbol{h}_{m_i,t}^i\right)$ and $\boldsymbol{h}_t$ for the vector $\left(\boldsymbol{h}_t^0 \quad \cdots \quad \boldsymbol{h}_t^n\right)$. This shows that the remaining safety budget and it can easily be tracked in order to assess the constraint satisfaction. Similarly, to the single agent case the recursive update is

$$\boldsymbol{h}_{j,t+1}^i = \left(\boldsymbol{h}_{j,t}^i - C_j^i\left(\boldsymbol{s}_t, \boldsymbol{a}_t\right)\right) / \gamma_c. \tag{4}$$

Then, during the policy training, we can reshape the reward by using the vector of hazard values $\boldsymbol{h_t}$:

$$\widetilde{\mathcal{R}}(\widetilde{s}_t, \boldsymbol{a}_t) = \begin{cases} \mathcal{R}(s_t, \boldsymbol{a}_t) & \forall i : \min_j \boldsymbol{h}_{j,t}^i \geq 0 \\ k & \exists i : \min_j \boldsymbol{h}_{j,t}^i < 0. \end{cases} \tag{5}$$

In the CTDE setting, according to equation 5, the algorithms will firstly find the minimum hazard value for each agent in the multiple constraints hazard vector $\min_j \boldsymbol{h}_{j,t}^i$. Then, suppose there exists one hazard value less than 0. In that case, this agent's trajectory already violates the adding constraints, so all agents will get an unsafe reward $k$ to avoid the same unsafe situation in the next time-step. Finally, we can formalise the problem to

$$\max_{\pi} \hat{J} \triangleq \widetilde{J}_{\mathcal{R}}(\pi) \triangleq \mathbb{E}_{\widetilde{s}_0, \boldsymbol{a}_0, \ldots} \left[\sum_{t=0}^{\infty} \gamma_r^t \widetilde{\mathcal{R}}(\widetilde{s}_t, \boldsymbol{a}_t)\right],$$
$$\text{s.t. } \widetilde{s}_t = s_t \times \boldsymbol{h}_{j,t}^i, s_0 \sim \rho^0(s_0), \boldsymbol{h}_{j,0}^i = b_j^i, a_t^i \sim \pi_i(\cdot \mid \widetilde{s}_t), \tag{6}$$
$$\widetilde{s}_{t+1} \sim \widetilde{\mathcal{P}}\left(\cdot \mid \widetilde{s}_t, a_t^1, \ldots, a_t^n\right), \boldsymbol{h}_{j,t+1}^i = \left(\boldsymbol{h}_{j,t}^i - \sum_{j=1}^{m^i} C_j^i\left(s_t, a_t^i\right)\right) / \gamma_c.$$

Note that the hazard values can be considered part of the transition dynamics, so that it can stay the Markov feature. Further, the overall problem is the standard CTDE formulation, which enables the plug-and-play feature.

## 4.3 CAMA IN IL

The CAMA in IL has the same process to calculate the hazard value vector, following equation 3 and updating the rule by equation 4. However, each agent can only observe its hazard value vector. Therefore, we reshaped the individual reward functions as follows,

$$\widetilde{r}_t^i = \widetilde{\mathcal{R}}_n(\widetilde{s}_t^i, a_t^i) = \begin{cases} \mathcal{R}^i(s_t^i, a_t^i) & \min_j \boldsymbol{h}_{j,t}^i \geq 0 \\ k & \min_j \boldsymbol{h}_{j,t}^i < 0. \end{cases} \tag{7}$$

Where for CAMA in the IL setting, considering each agent has its local critic and actor, and the reshaped reward function is the part that each agent interacts with the environment. Note that the reward is replaced by the negative number $k$ when this agent's minimum local hazard value is less than 0. The benefit is that it follows the original intention of the IL setting, which is to treat the surrounding agents as environment so that neighbors' violations do not affect their rewards.

---

**Algorithm 1:** **C**onstraint **A**ugmented **M**ulti-**A**gent Reinforcement Learning (CAMA)

---

1: **for** $N_{episodes}$ **do**
2:     Initialize the scenario and the agent
3:     Reset state $s_0$, Initialize a random process $\mathcal{P}_0$ for action exploration
4:     **for** $t = 0, 1, \ldots$ **do**
5:       **for** each agent **do**
6:         Select the action $a_i$ under the current policy $\pi_i$ and its exploration.
7:         Store those in the buffer.
8:       **end for**
9:       Executes the actions $\boldsymbol{a}$ and calculate the new state $s'$
10:      Sample the hazard value vector $\boldsymbol{h}_t$ and augmented into the state $s_t$ // Eq 4
11:      Get the new reshaped Reward $\widetilde{\mathcal{R}}(\widetilde{s}_t, \boldsymbol{a}_t)$ // Choose Eq 5 or 7 depending on the game setting.
12:      Updating policies $\boldsymbol{\pi}$, to maximise $J_{\mathcal{R}}(\boldsymbol{\pi})$ through the MARL algorithm //Eq 6
13:    **end for**
14: **end for**

---

Therefore, we can present CAMA's framework and internal logic through the algorithms.1. For both CTDE and IL settings, CAMA will follow the general update process in steps 1-9 until all agents' following actions and their ground truth rewards are obtained. Then, in steps 10 to 12, the hazard values and reshaped rewards are updated according to the two settings discussed above. In the end, update the policy by using new reshaped reward and augmented hazard value to state to wait next time-steps training. This flexible framework enables CAMA to improve the safety without changing the combined algorithm.

## 5 Experimental Results

We compare CAMA with the Safety MAMujoco baseline (Gu et al., 2021) and the GymCompete baseline (Bansal et al., 2017). We also provide ablation studies on unsafe reward and safety budget components.

**General Implementation.** To use the CAMA framework, we only need to calculate the hazard value, reshape the rewards, and augment them into their trajectory. Thus, there is no need to modify the non-safe MARL algorithm itself. During the training, the reset and step function will augment the hazard value and reshape the new cost function according to equation.5. We combine the CAMA with HAPPO (Kuba et al., 2022), MAPPO (Yu et al., 2021) and IPPO (de Witt et al., 2020) to create CAMA-HAPPO, and CAMA-MAPPO for CTDE setting and CAMA-IPPO for IL setting. For the baseline comparisons, we compare our method with the MACPO, MAPPO-Lagrange (Gu et al., 2021), and those originally non-safe algorithms itself. We used the hyperparameters that reported the best performance of the baseline algorithms. More details of the hyperparameters setting can be found in Appendix E.

### 5.1 CTDE Experiment

In this subsection, we mainly introduce the performance of CAMA in the cooperative MARL game. In cooperative tasks, different agents will share a common goal and maximize team reward.

**Joint Reward Environments setup.** We demonstrate the benefit and limitations of our method on SMAMuJoCo (Gu et al., 2021), which is a safety-aware extension of Mujoco (Todorov et al., 2012) designed for safe MARL research. The main three scenarios we used are Ants, Half-Cheetahs, and ManyAnt (schematically depicted in Figure.1). The number behind the robot name is the number of agents and the parts each agent controls, like Ant 2x4, which means there are two agents and control two neighbour legs together (each lag has two parts), and Ant 2x4d indicate the diagonal two legs

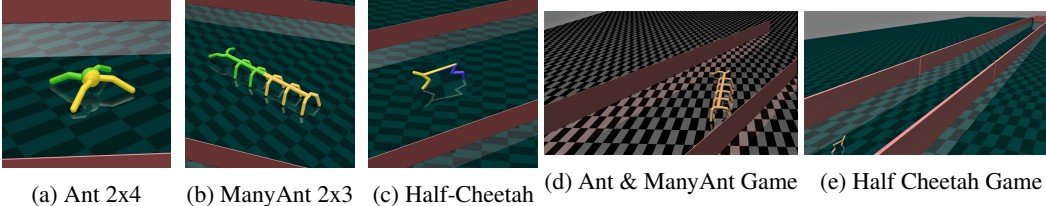

(a) Ant 2x4    (b) ManyAnt 2x3    (c) Half-Cheetah    (d) Ant & ManyAnt Game    (e) Half Cheetah Game

Figure 1: Cooperate Scenarios: Panels a to c: The view of robots. Multi-agents control the body by different colour parts. Panels d: The Ant and ManyAnt scenario with overthrow-able walls, Panels e: The Half Cheetah scenario with a moving heading obstacle and overthrow-able walls.

control by two agents. All the task aims to let agents jointly learn the manipulation of the robot while avoiding crashing dark red walls into an unsafe area.

For the environment costs design, we follow the previous work by (Gu et al., 2021) , more details listed in Appendix B, every agent will receive cost from the environment, and we calculate the average and maximum for performance comparisons. We all run in 5 different seeds for all following experiments and evaluate the result using the saved intermediate policies for every agent. Please note that, in each episode, for each agent, the system will present two types of rewards. One is *true reward* $r_t$ based on the original task reward, we sum all agents' true reward as the team reward (titled Average Episode Cost in the figure). The other one is *reshaped reward* $\widetilde{r}_t$ based on *hazard value* $h^i_{j,t}$, the policy receives this as a feedback from the environment for training.

In addition, two main hyper-parameters can be tuned in CAMA: the safety budget $b$ and unsafe reward $k$. In order to have a fair comparison, we set the safety budget equal $50$ and the unsafe reward to equal to $k_t = -\mathcal{R}(s_t, a_t)$ for all tasks, for more setting discussion and its ablation test follow in sec.5.3.

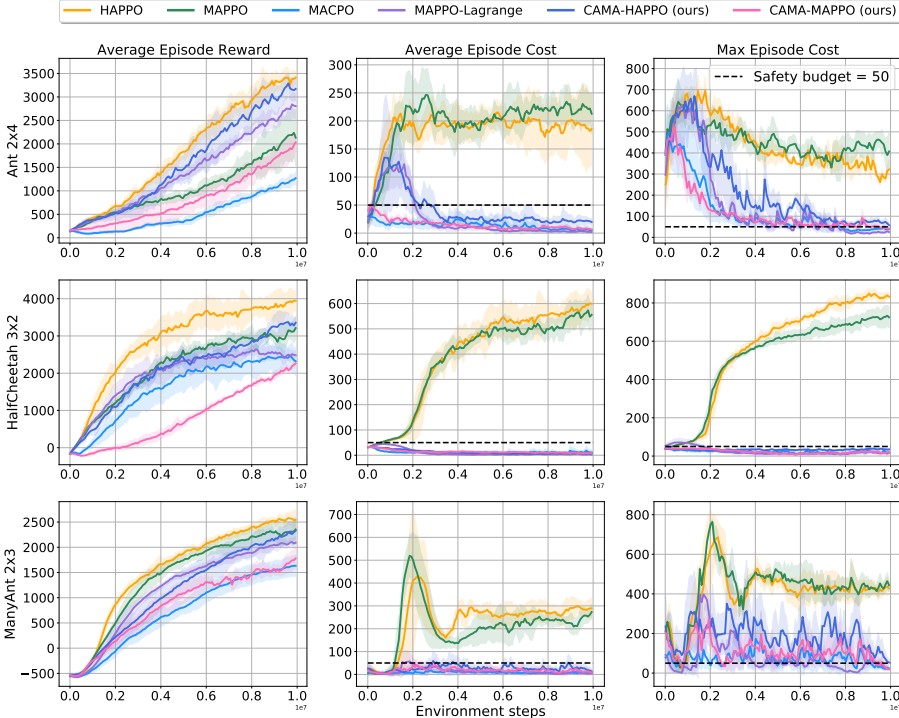

Figure 2: **CAMA in SMAMuJoCo with joint reward.** Performance comparisons between CAMA-MAPPO, CAMA-HAPPO with their rivals in the CTDE settings. Our approach converges average and max costs to a priori safety budgets. Easily enable non-safe MARL algorithms that meet constraints.

**Main Result in CTDE.** Figure 2 demonstrate the performance comparisons in the CTDE setting, including tasks of Ant 2x4d, Half-Cheetah 3x2, ManyAnt 2x3 in terms of Average Episode Reward, Average Episode Cost and Max Episode Cost. The dotted line are the safety budget set to 50 for all tasks. In terms of cost, compared to the original non-safe algorithms MAPPO and HAPPO, the CAMA version ensures that the average cost stays close to zero and the Max cost converges to the safety budget. Regarding reward, the CAMA-HAPPO consistently outperforms MACPO and MAPPO-L but performs its original non-safe algorithm. The CAMA-MAPPO can hold similar performance compared to MACPO but lower than MAPPO and MAPPO-Lagrange. The figures show that our approach significantly improved the safety of the algorithm with only a slight reduction in the efficiency of the reward.

## 5.2 IL EXPERIMENT

To demonstrate the plug-and-play nature of CAMA, and its generality, we also test its performance in both cooperative and competitive IL games.

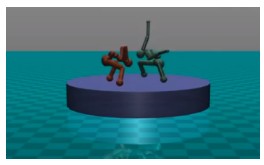 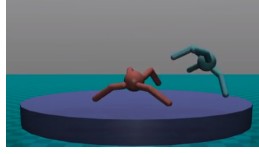

(a) Sumo-Human       (b) Sumo-Ant

Figure 3: Gym Compete Sumo Scenarios: Agents compete in a circular arena, each with the goal of knocking another agent to the ground or pushing them out of the ring.

**Sum Independent Reward Environments setup. (Cooperative)** The setting of the environment is similar to SMAMuJoCo CTDE. Each agent has an independent policy (rather than each part of the robot), but updating the policy refers to its local state and reward. To better show the whole system's performance, we will calculate the sum of the independent rewards and costs by each agent. Note that those measure values are only used for analysis.

**Zero-sum Game Environments setup. (Competitive)** We involve the CAMA in Gym Compete (Bansal et al., 2017) to demonstrate performance in a competitive setting. The tested scene is shown in Figure 3, which called Sumo. We mainly tested two different robots, namely Ant and Humans, and each color represents an agent (Note that, unlike SMAMuJoCo, the robot is no longer divided into multiple parts). The Sumo-Human environment will be more difficult than the Sumo-Ant, since the initial state dimensions of the two robots are different ($D_{Human} = 24$; $D_{Ant} = 15$), please refer to Appendix C for details.

In a sumo competition, each player must stay on the field within 5m. And try to push your opponent out of bounds as much as possible to win. For each agent, the closer the edge, the more dangerous it is. Therefore, CAMA can convert "not falling" into a condition that needs to be met. The direct idea is to use the distance from the agent to the center as a reference. We set the safe radius to 2.5m and the safety budget to 1.0. The environment will give the agent a 0.1 cost when out of range. Similar to the previous implementation, we use five random seeds for testing and calculate the average winning rate and episode cost for analysing

**Main Result in IL.** Figure 4 demonstrates the results for the cooperative IL setting, which includes Ant 2x4d, and Half-Cheetah 3x2 tasks. In terms of cost, compared to the original IPPO, the CAMA-IPPO ensures that the average cost stays close to zero, and the Max cost converges to the safety budget in both tasks. In terms of reward, CAMA-IPPO not only have a similar performance to IPPO in half- cheetah, but it also outperformed in the ant task. The reason probably stems from the reward setting of the environment, as the reward includes a portion of the cost penalty, leading to a relatively higher reward for the safe algorithm. All in all, the figures show that our approach results in a significant improvement in constraint satisfaction and limited enhancements in rewards.

Figure 5 shows the CAMA player verse the original IPPO player in the competitive IL game. In sumo-ant, CAMA can significantly improve the win rate while converging to the lower cost. We also found that CAMA tends to be a counter-attack player, which tries to attack the opponent while ensuring itself in a safe area. We speculate that CAMA will choose to avoid the attack due to the heavier penalties for violating the safety constraints in the later stage. On the contrary, the original IPPO attacked desperately and easily rushed to the edge. In the more challenging scenario sumo-human, we found that although CAMA will still be a low win rate at the beginning, it will eventually tend to converge to a state of equilibrium (the winning rate of both sides tends to be

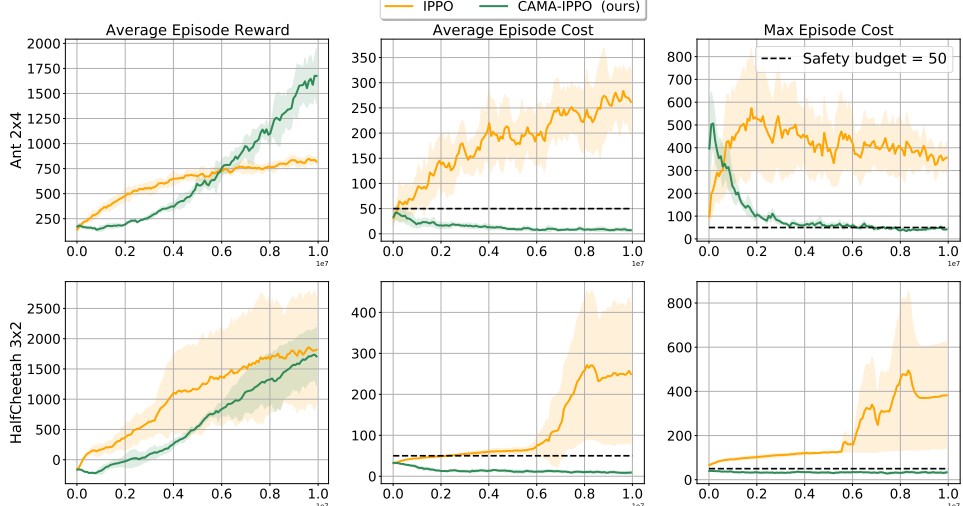

Figure 4: **CAMA in SMAMuJoCo with independent reward.** Performance comparisons between CAMA-IPPO and IPPO in the IL settings. With the benefit of satisfying the constraints, it has been possible to exceed the performance of IPPO in the both scenario.

50%). According to the cost figure, in higher dimensional and more complex situations, CAMA is hard to learn attack strategies and instead aims to avoid attack and stalemate with the opponent. This strategy is reminiscent of the attack strategy proposed by Gleave et al. (2019) in performance. Nonetheless, there is an essential difference between the two: CAMA players are defences chosen under the premise of satisfying constraints and still have the possibility of anti-policy learning. At the same time, adversarial strategies change the victim through the natural behaviour of the attacker state. Interestingly, the two perform similarly, and it may be a potential research direction to guide the agent to learn aggressive strategies by adjusting constraints.

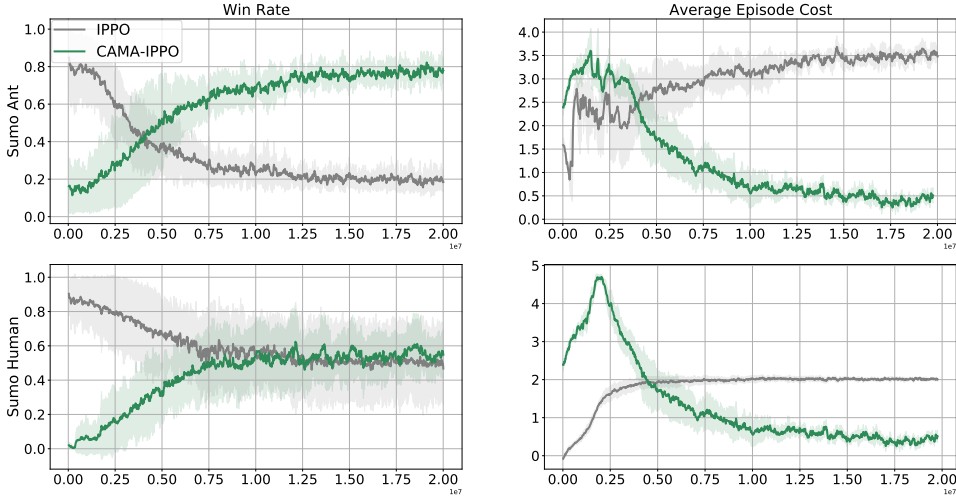

Figure 5: **CAMA in GymCompete with independent reward.** Performance comparisons for CAMA-IPPO verse IPPO in a competitive game. In both sumo-ant and sumo-human, CAMA based IPPO have higher win rate and lower average cost.

## 5.3 ABLATION STUDIES

**The Dynamic Unsafe Reward in CTDE.** In order to facilitate the calculation, the unsafe reward is usually defined as a very large negative number, which is feasible in the IL setting. However, in the CTDE environment, substantial unsafe rewards will lead to the problem of low learning efficiency and non-convergence, and we discuss the reasons in detail in Appendix D. Intuitively, in the setting of CTDE, all other agents are severely penalized for a violation by one agent,which affects the policy exploration of the safe agent. Therefore, we changed the unsafe reward from fixed to dynamic. We let the unsafety reward become proportional to the getting reward at the time step, as $k_t = -\mathcal{R}(s_t, \boldsymbol{a}_t)$. After testing, we found that when one or more agents violate the constraints in multi-agent to some extent, this dynamic unsafe reward can reduce the impact on the rewards of other agents. Therefore, we do not need to tune this as a hyper-parameter anymore.

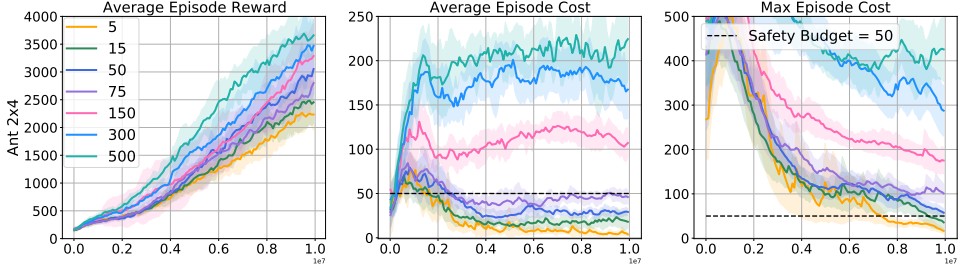

Figure 6: Ablation Result in Safety Budget , unsafe reward $k_t = -\mathcal{R}(s_t, a_t)$

**Safety Budget.** As the initial value of the $\boldsymbol{h}$, we can generalize the different budgets before or after the training process. We progressively selected seven different budgets in the Ant2x4 joint reward game, ranging from very low initial constraints **"5"** to almost no constraints at the beginning **"500"**. The results are shown in Figure 6, and the final convergence details can be found in Appendix D table 1. It can be found that under the same setting of the unsafe reward $-\mathcal{R}(s_t, a_t)$, all algorithms have high efficiency in average team reward, and the maximum cost can almost satisfy the constraints in the end. We set the safety budget as $50$ in all SMAMuJoco experiments and $1.0$ in all Gym Compete Sumo experiments, and other details of the settings can be viewed in appendix E.

**Ablation on CAMA components.** Figure.7 demonstrates the performance after removing two critical components in CAMA. Removing constraint enhancement leads to lower learning efficiency but still keeps the safety constraints satisfied. It appears that the constraints and its budget value still be involved during reward calculation, thus being forced to converge to the conservative policy. Conversely, removing reward reshaping has better rewards but no longer satisfies the constraints. While in contrast, the cost is still lower than the original algorithm, which is related to the augmentation of the hazard value. Overall, this experiment demonstrates that these two components play a crucial role in CAMA.

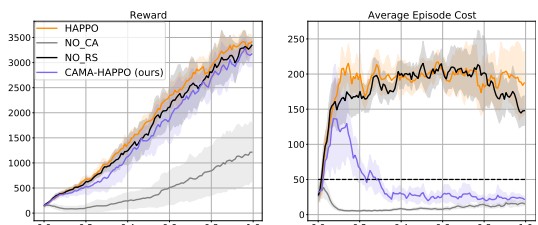

Figure 7: The ablation on CAMA components in joint reward Ant2x4d scenario. "No CA" stands for without constraint augmentation, "No RS" stands for without reward reshaping.

## 6 CONCLUSION

This work proposed CAMA, which is a plug-and-play framework of Safe MARL using constraint augmentation. The proposed framework improve safety of existing multi-agent learning algorithms without pre-training and complex implementations. Furthermore, our framework is compatible with different MARL settings and interface seamlessly with both cooperative and competitive settings. Empirical results on cooperative and competitive tasks demonstrate the superiority of the CAMA-based algorithms in improving safety and maintaining the same level of reward performance.

REPRODUCIBILITY STATEMENT

The details of the experiment settings are provided in Section 5.1 and 5.2. The hyparameter for the proposed algorithm and baselines are in Appendix E. The pseuco code is summarized in Algorithm 1. The ablation test detail are provided in Section 5.3 and Appendix D. We provide detailed proof of theoretical analysis in Appendix A. A more detailed description and implementation setting can be found in Appendix B for SMAMujoco and Appendix C for Gym Compete.

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

# A    THEORETICAL ANALYSIS

In this paper, we extend the theoretical analysis for the single agent case Sootla et al. (2022a) to the multi-agent case. The authors of Sootla et al. (2022a) define the problem in the following form,

$$\max_{\pi} \mathbb{E}_{s_0, \boldsymbol{a}_0, \dots} J_{\mathcal{R}}(\pi), \boldsymbol{a}_i \sim \pi(\cdot | s_i) \tag{8a}$$

$$\text{s. t.: } J_{\mathcal{L}}(\pi) \leq b, \ \ \text{W.P. 1} \tag{8b}$$

where $b$ is the safety budget and  W.P. 1 stands for with probability one. In this setting the goal is to find the policy (8a), which maximizes the task reward while making the cost less than a pre-defined safety budget $b$ with probability one (8b). To facilitate training, they actually solved the following problem,

$$\max_{\pi} \mathbb{E}_{s_0, \boldsymbol{a}_0, \dots} J_{\mathcal{R}}^n, \boldsymbol{a}_i \sim \pi(s_i)$$

$$J_{\mathcal{R}}^n \triangleq \sum_{t=0}^{\infty} \gamma_c^t \widetilde{\mathcal{R}}_n(\widetilde{s}_t, a_t) \tag{9}$$

Their key result was to show that with $n \to -\infty$ the cost $J_{\mathcal{R}}^n$ converges monotonically to $J_{\mathcal{R}}$ — the optimal cost after solving Problem 8. The authors of Sootla et al. (2022a) invoked classical results for optimal control to prove this assertion (Hernández-Lerma & Lasserre, 2012). They made the following three assumptions:

**Assumption 2** *The functions $\widetilde{\mathcal{R}}(\widetilde{s}, a)$ are bounded, measurable, nonnegative, and upper semi-continuous on $\widetilde{\mathcal{S}} \times \mathcal{A}$*

**Assumption 3** $\mathcal{A}$ *is compact;*

**Assumption 4** *The transition $\widetilde{\mathcal{P}}$ is weakly continuous on $\widetilde{\mathcal{S}} \times \mathcal{A}$, i.e., for any continuous and bounded function $u$ on $\widetilde{\mathcal{S}}$ the map $(\widetilde{s}, a) \to \int_{\widetilde{\mathcal{S}}} u(x, y) \mathcal{P}(dx, dy | \widetilde{s}, a)$ is continuous.*

Note that in the results in Sections 4.2 and 4.3 in (Hernández-Lerma & Lasserre, 2012), the authors used milder assumptions than Assumptions A1 and A2. However, Sootla et al have found that Assumptions A1 and A2 are often met in RL applications, and hence there was no need to complicate the presentation. Assumption A3 is also mild in the RL setting as discussed in (Arapostathis et al., 1993) in Section 2.4. Indeed, this assumption hols when the transition $\mathcal{P}$ is a Gaussian distribution with continuous mean and variance. Now, consider the problem with the objective $J_{\text{task}}^n$ in Equation 9 satisfying the assumptions above. Then

**Theorem 1** *For any finite unsafe reward $k$, the Bellman equation is satisfied, i.e., there exists a function $V_n^*(\widetilde{s})$ such that*

$$V_n^*(\widetilde{s}) = \max_{a \in \mathcal{A}} \left( \widetilde{\mathcal{R}}_n(\widetilde{s}, a) + \gamma_r \mathbb{E}_{\widetilde{s}'} V_n^*(\widetilde{s}') \right),$$

*where $\widetilde{s}' \sim \widetilde{\mathcal{P}}(\cdot | \widetilde{s}, a)$. Furthermore, the optimal policy solving $J_{\mathcal{R}}^n$ has the representation $\boldsymbol{a} \sim \pi_n^*(\cdot | \widetilde{s})$.*

**Theorem 2** *The optimal value functions $V_n^*$ for $J_{\mathcal{R}}^n$ converge monotonically to $V_{\infty}^*$ — the optimal value function for $J_{\mathcal{R}}^{\infty}$.*

**Theorem 3** *Suppose there exists an optimal policy $\pi^*(\cdot | \widetilde{s}_t)$ solving Equation 9 for the objective $J_{\mathcal{R}}^{\infty}$ with a finite cost, then $\pi^*(\cdot | \widetilde{s}_t)$ is an optimal policy for Equation 8.*

Now we discuss this results from MARL perspective. In the IL setting, the independent learners consider other agents as the environment. Hence intuitively if these independent learners satisfy the Bellman equation in the unconstrained case, then Theorem 1 would show that safe IL would also satisfy the Bellman equation in the constrained case. Therefore, we can use critic-based methods for the safe version and guarantee their convergence under those assumptions, at the same time each agent's optimal policy would be Markovian.

In the CTDE setting, Assumption A1 can be directly extended because the reward function will evaluate all agents and return the team reward by summing, in case of upper semi-continuous, $J_{\mathcal{R}}^{n,i} \to J_{\mathcal{R}}^i$. According to (Kuba et al., 2022), in some CTDE cases we also need the following regularity assumptions, which in particular, guarantees that the set of policies is closed and bounded:

**Assumption 5** *There exists $\theta \in \mathbb{R}$, such that $0 < \theta \ll 1$, and for every agent $i \in \mathcal{N}$, the policy space $\Pi^i$ is $\theta$-soft, which means for every $\pi^i \in \Pi^i, s \in \mathcal{S}$, and $a^i \in \mathcal{A}^i$, we have $\pi^i\left(a^i \mid s\right) \geq \theta$.*

Indeed, according to Lemma 3 (Kuba et al., 2022), let $\left(\pi_t^i\right)_{t=0}^{\infty}$ be a convergent sequence of policies of agent $i$. Then according to the Assumption A4, for any $k \in \mathbb{N}, s \in \mathcal{S}$, and $a^i \in \mathcal{A}^i, \pi_t^i\left(a^i \mid s\right) \geq \theta$, hence, $\pi^i\left(a^i \mid s\right) = \lim_{t\to\infty} \pi_t^i\left(a^i \mid s\right) \geq \lim_{t\to\infty} \theta \geq \theta$. Furthermore, since sum of probability for actions sampled from policy is one, so that $\left|\pi^i\left(a^i \mid s\right)\right| \leq 1 \to \left\|\pi^i\right\|_{\max} \leq 1$, which proved the boundedness of the set of polices.

Finally, let us discuss Assumption A3 in CTDE. Although, the CTDE setting increases the dimension of the hazard value according to the cost equation, each agent will eventually return the smallest and continuous hazard value for augmentation with the current state, so the assumption is still true. Therefore, we can extend Theorem 1 to both CTDE and IL by satisfying the assumptions A1-A4.

## B SMAMuJoCo Environments:

Safe Multi-Agent MuJoCo (SMAMuJoCo) (Gu et al., 2021) is an extension of MuJoCo (Todorov et al., 2012). In particular, while retaining the background environment, proxy, physics simulator and reward functions, adding collision obstacles in the environment, such as fixed walls and moving baffles, can be touched and knocked over when agents valid the constraint. During the training process, when the agent produces unsafe behaviours like crash, the environment will generate costs according to the settings of different scenarios. This section will introduce the cost settings for each task used in this paper.

### ManyAnt & Ant

ManyAnt environment (8a) has a linear corridor with a width of 9m. Ants (8b) environment has three sections of polyline corridors with an angle of 30 degrees, and the width between the two walls is 10m. In both environments, which returns cost if the distance between the agent and the wall is less than 1.8m or the agents falls over. Therefore,

$$
c_t^i = \begin{cases} 0, \ \textbf{(Safe)}, & \text{If} \quad 0.2 \leq z_{t+1}^{\text{agent}} \leq 1.0 \text{ and } \left\|x_{t+1}^{\text{agent}} - x^{\text{wall}}\right\|_2 \geq 1.8, \\ 1, \ \textbf{(Unsafe)}, & \text{Otherwise .} \end{cases}
$$

where $z_{t+1}^{\text{agent}}$ is the agents torso's $z$-coordinate, and $x_{t+1}^{\text{agent}}$ is the agents torso's $x$-coordinate. For each time step, the system need to calculate the distance between agents torso with $x^{\text{wall}}$, which is the $x$-coordinate of the wall.

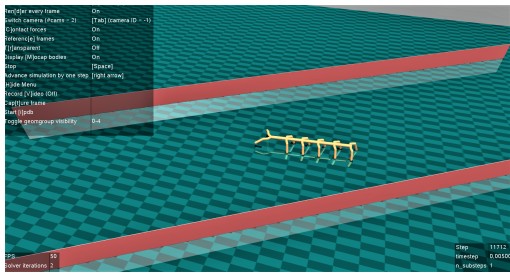

(a) ManyAnt Scenario

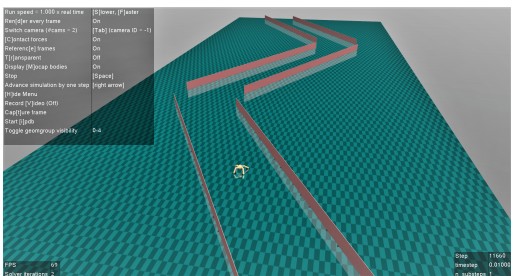

(b) Ant Scenario

### Half-Cheetah

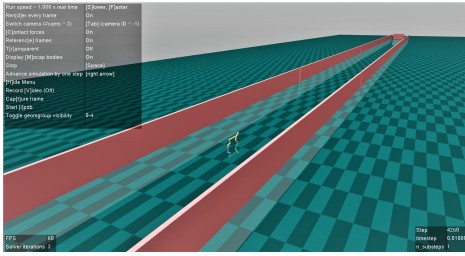

Figure 9: Ant 2x4

In this mission, the agent moves in a special corridor, the fixed walls on both sides no longer incur costs and do not get knocked down, but which will restricts agents' movement. Newly added is a moving baffle in the corridor, which incurs a cost of 1 if the distance between the agent and the baffle is less than 9 m, then the cost $c_t$ can be demonstrated as,

$$c_t = \begin{cases} 0, & \textbf{(Safe)}, \text{ If } \left\| \boldsymbol{y}_{t+1}^{\text{agent}} - \boldsymbol{y}^{\text{baffle}} \right\|_2 \geq 9 \\ 1, & \textbf{(Unsafe)}, \text{ otherwise }. \end{cases}$$

where $\boldsymbol{y}_{t+1}^{\text{agent}}$ is the $y$-coordinate of the agents torso, and $\boldsymbol{y}^{\text{baffle}}$ is the $y$-coordinate of the baffle.

## C   COMPETITIVE ENVIRONMENTS:

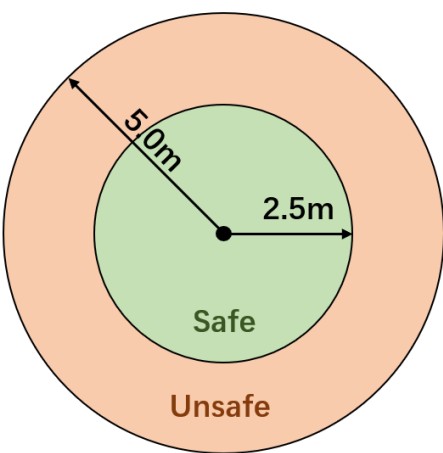

Figure 10: Sumo area

Gym Compete (Bansal et al., 2017) Sumo allows agents to compete in a 1-vs-1 system following standard sumo rules. There are two types of agents, Ant and Humans, which have different anatomical structures. During the game, The players' positions are randomly initialized, and each agent observes its own and its opponent's positions, its own joint angles, the corresponding speed, and the force applied to its own body (i.e., the equivalent of touch). The action space is continuous. Figure.10 describes the safe area in the arena, that is, a circle with a radius of **2.5m**. The cost given by the environment is set as follows,

$$c_t = \begin{cases} 0, & \textbf{(Safe)}, \text{ If } \left\| \boldsymbol{x}_{t+1}^{\text{agent}} - \boldsymbol{x}^{\text{center}} \right\|_2 \leq 2.5 \\ 0.1, & \textbf{(Unsafe)}, \text{ otherwise }. \end{cases}$$

where $\boldsymbol{x}_{t+1}^{\text{agent}}$ is the $x$-coordinate of the agents torso, and $\boldsymbol{x}^{\text{center}}$ is the $x$-coordinate of the center of circle.

# D   ABLATION EXPERIMENTS

| $b$ | Average **R** | Max **C** | $k$ | Average **R** | Max **C** |
|---|---|---|---|---|---|
| 0 | $[1709.78 \pm 123.66]$ | $[3.12 \pm 0.4]$ | $-\mathcal{R}(s_t, a_t)$ | **$[3299.01 \pm 518.19]$** | $[44.0 \pm 6.41]$ |
| 5 | $[2233.00 \pm 429.62]$ | $[6.2 \pm 0.49]$ | -0 | $[2959.34 \pm 357.12]$ | $[75.0 \pm 13.295]$ |
| 15 | $[2508.28 \pm 390.03]$ | $[14.7 \pm 0.40]$ | -5 | $[1964.27 \pm 311.90]$ | $[45.0 \pm 9.559]$ |
| **50** | **$[3299.01 \pm 518.19]$** | $[44.0 \pm 6.41]$ | -10 | $[1447.00 \pm 364.32]$ | $[41.0 \pm 12.77]$ |
| 75 | $[2923.27 \pm 180.27]$ | $[82 \pm 11.84]$ | -20 | $[1170.36 \pm 371.57]$ | $[69.33 \pm 17.72]$ |
| 150 | $[3376.28 \pm 603.48]$ | $[169 \pm 24.07]$ | -50 | $[522.46 \pm 195.17]$ | $[79.0 \pm 19.05]$ |
| 300 | $[3627.82 \pm 576.99]$ | $[273 \pm 65.65]$ | -100 | $[364.72 \pm 51.68]$ | $[76.0 \pm 7.79]$ |
| 500 | $[3737.58 \pm 97.50]$ | $[416 \pm 89.43]$ | -1000 | $[31.34 \pm 18.55]$ | $[48.0 \pm 12.86]$ |

Table 1: Ablation test in Safety Budget $b$     Table 2: Ablation test in Unsafe Reward $k$

**Further Ablation experiments for unsafe rewards.** In previous experiments, we know that for different scenarios or Markov game settings, we should use different unsafe rewards. Here, we test the performance of both fixed and dynamic values of $k$ under a safety budget of 50, and present the results of one million training evaluations in Table 2. It was found that when the fixed $k$ is reduced, the policy learning is inefficient or even unable to converge. This result significantly differs from the single-agent setting. We consider this a unique situation worth discussing in the multi-agent environment. Under the CTDE setting, when one of the agents violates the constraint, the team reward becomes the unsafe reward times the number of agents. Then it means that the reshaped rewards will be much smaller than the real rewards given to the agents by the environment. As a result, the centralized critic will ignore the general policy exploration of agents who do not violate the constraints, resulting in low training efficiency. At the same time, too small $k$ can also lead to other numerical issues in training. Therefore, for this situation, it is necessary to balance the relationship between environmental reward and unsafe reward to improve learning efficiency while satisfying the constraints. In order to do that, an intuitive idea is to set a dynamic $k$ that grows larger over the time, and eventually reaching negative infinity. Thus, we set $k = -\mathcal{R}(s_t, a_t)$. The advantage is that the reward is small at the beginning, so the corresponding penalty is also small, which encouraging agents to explore the policy. With this dynamic setup, we find that CAMA can better adapt to difficult tasks and tends to converge (Table 2, first row). Therefore, in CTDE and IL, we set $k = -\mathcal{R}(s_t, a_t)$.

Through the detailed convergence comparison in Table 2, we finally determined to use $-0.2$ as an unsafe reward in the Ant environment to ensure the stability of learning.

# E   HYPER-PARAMETERS SETTING FOR EXPERIMENTS

The following four tables describe all the hyperparameters used in practice, respectively. For CAMA-base MARL algorithms, besides the *Safety Budget and Unsafe Reward*, other parameters are the same as the occupation, in order to have the fair comparison in experiments. Notably, these used hyperparameters are all cited from their best performing experiments in the original paper.

| hyperparameters | value | hyperparameters | value | hyperparameters | value |
|---|---|---|---|---|---|
| critic lr | 5e-3 | optimizer | Adam | num mini-batch | 40 |
| gamma($\gamma_c, \gamma_r$) | 0.99 | optim eps | 1e-5 | batch size | 16000 |
| gain | 0.01 | hidden layer | 1 | training threads | 4 |
| std y coef | 0.5 | actor network | mlp | rollout threads | 16 |
| std x coef | 1 | eval episodes | 32 | episode length | 1000 |
| activation | ReLU | hidden layer dim | 64 | max grad norm | 10 |

Table 3: Common hyperparameters used for **CAMA-HAPPO, CAMA-MAPPO, CAMA-IPPO**, MAPPO-Lagrangian, MAPPO, HAPPO, IPPO, and MACPO in the Safe Multi-Agent MuJoCo ManyAnt, Ant, Half-Cheetah, and Gym Compete Sumo-Ant, Sumo-Human

| Algorithms | CAMA-HAPPO | HAPPO | CAMA-MAPPO | MAPPO | MACPO |
|---|---|---|---|---|---|
| actor lr | 9e-5 | 9e-5 | 9e-5 | 9e-5 | / |
| ppo epoch | 5 | 5 | 5 | 5 | / |
| kl-threshold | / | / | / | / | 0.0065 |
| ppo-clip | 0.2 | 0.2 | 0.2 | 0.2 | / |
| fraction | / | / | / | / | 0.5 |
| fraction coef | / | / | / | / | 0.27 |
| Lagrangian coef | / | / | / | / | / |
| Lagrangian lr | / | / | / | / | / |

Table 4: Different hyperparameters used for **CAMA-HAPPO**, HAPPO, **CAMA-MAPPO**, MAPPO, and MACPO in the SMAMuJoCo domains.

| Algorithms | CAMA-IPPO | IPPO | MAPPO-Lagrangian |
|---|---|---|---|
| actor lr | 9e-5 | 9e-5 | 9e-5 |
| ppo epoch | 5 | 5 | 5 |
| kl-threshold | / | / | / |
| ppo-clip | 0.2 | 0.2 | 0.2 |
| fraction | / | / | / |
| fraction coef | / | / | / |
| Lagrangian coef | / | / | 0.78 |
| Lagrangian lr | / | / | 1e-3 |

Table 5: Different hyperparameters used for **CAMA-IPPO**, IPPO, MAPPO-Lagrangian in the both SMAMuJoCo and Gym Compete domains.

| task | Safety Budget | Unsafe Reward |
|---|---|---|
| Ant(2x4d) | 50 | $-\mathcal{R}(s_t, a_t)$ |
| HalfCheetah(3x2) | 40 | $-\mathcal{R}(s_t, a_t)$ |
| ManyAnt(2x3) | 30 | $-\mathcal{R}(s_t, a_t)$ |

Table 6: The new two hyper-parameters *Safety Budget* and *Unsafe Reward* used for CAMA-HAPPO, CAMA-MAPPO, CAMA-IPPO in the SMAMuJoCo domains.

| task | Safety Budget | Unsafe Reward |
|---|---|---|
| Sumo-Ant | 1.0 | $-\mathcal{R}(s_t, a_t)$ |
| Sumo-Human | 1.0 | $-\mathcal{R}(s_t, a_t)$ |

Table 7: The new two hyper-parameters *Safety Budget* and *Unsafe Reward* used for CAMA-IPPO in the Gym Compete domains.

