# OpenReview forum: "CAMA: A New Framework for Safe Multi-Agent Reinforcement Learning  Using Constraint Augmentation"
_ICLR.cc/2023/Conference — Submitted to ICLR 2023_

### Official Review · Reviewer_nfFv · 2022-10-23

**Confidence:** 4
**Correctness:** 3
**Technical Novelty And Significance:** 2
**Empirical Novelty And Significance:** 2
**Recommendation:** 5

**Clarity, Quality, Novelty And Reproducibility:**

Paper is not well written (see details). The idea is incremental from the novelty standpoint. Authors provided the full list of their parameters in the appendix. Although I could not find the separate discount values they used. does gamma = .99 correspond to \gamme_c or \gamma_r or both?

Details:
- "Although these methods satisfy the constraints to a certain extent, a part of the reward performance is still sacrificed" -> Whenever you want to enforce safety you have to sacrifice part of performance.
- "such asElSayed-Aly et al." -> Add space before citation
- "Recent works, such like" -> Recent works like
- Excessive use of ";"
- Subscript j and m^i is not defined in the definition of C^i_j
- Why pi maps to [0,\infty] instead of [0,1] shouldn't the output be a probability?
- [Notation] The R symbol used to represent the CMMDP and the one later defined as R_i are different. The relation is not officially defined. I believe R_i is the i_th element of R defined in the tuple.
- Is C^i = sum_j C^i_j? You have not defined it.
- a n-agent -> an n-agent
- "showed that an algorithm solving this problem actually solves a safe RL with probability one constraints." -> please add "under certain conditions."
- Why \tilde R_i is assumed to be non-negative?
- What is the motivation behind defining h in formula 4? Why divide by the constant gamma? Is it for simplicity of calculation?
- If k is negative, how do you ensure the assumption of \tilde{R_i} to be non-negative as defined in Definition 1?
- Why set k to -R(spa)? Wouldn't this avoid any negative penalty if R(s,a) = 0? I read the discussion in the appendix but not sure how the reasoning hold in border set of domains.
- I don't see CAMA-MACPO in figure 2. Typo? Did you mean CAMA-MAPPO?
- "More Harder" => More difficult
- "agent..": remove one dot
- "to ensure maximum exploration efficiency": What does it mean?
- Did you mention the your gamma_r and gamma_c for your experiments? Are both of them .99 given table 3 in the appendix?

**Strength And Weaknesses:**

Strengths
+ Bringing the constraint state augmentation into the multi-agent setting is appealing
+ The approach can be combined with any method, as the result of CAMA formulation is a MARL problem.

Weaknesses
- The writing is not great. Several improvements are needed (see details below)
- Certain definitions seems incorrect. For example why \tilde R_i is assumed to be non-negative in Definition 1 given that k is often negative? (see details below)
- The contribution seems incremental given the work of [Sootla et al. 2022].
- Experimental results don't show a major difference with MAPPO-Lagrange approach. In the literature review section authors mentioned: "Although these methods satisfy the constraints to a certain extent, a part of the reward performance is still sacrificed". This is also the case for their approach.


**Summary Of The Paper:**

Authors introduced the framework of Constraint Augmented Multi-Agent (CAMA) for solving constrained multi-agent reinforcement learning problems. Earlier Sootla et al. [2022] augmented the safety constrains into the cost function by defining a safety budget and panelizing the agent when crossing that budget. CAMA brings this trick into the multi-agent setting for both the centralized training and decentralized execution (CTDE), and individual learning (IL). The latter is pretty similar to the earlier work (single agent), while the CTDE required more expansion. Authors compared their algorithms against 4 approaches in 5 domains showing promising results.


**Summary Of The Review:**

The paper has great direction, yet I am not sure if the contributions of the paper is enough for a publication. Moreover the writing of the paper needs non-trivial amount of work.

---

### Official Review · Reviewer_CFTd · 2022-10-23

**Confidence:** 2
**Correctness:** 3
**Technical Novelty And Significance:** 3
**Empirical Novelty And Significance:** 2
**Recommendation:** 5

**Clarity, Quality, Novelty And Reproducibility:**

The paper is generally well-written.
However, it was initially difficult to understand what the actual contributions of this paper are versus the used techniques from the existing literature and how the new paper adds to the existing body of work.
This makes it difficult (for me) to judge the novelty specifically.

Details for the implementation of the method are given, although no code release is available. For an expert in MARL it might be able to reproduce the results of the paper.

**Strength And Weaknesses:**

The approach of CAMA is compatible with many MARL algorithms and does not require too big changes to be introduced.
Experiments show that the safety budget is better honoured than in other approaches, and that in some cases a better performance in terms of reward is achieved.
At the same time, the reward improvement is not consistent and in some environments CAMA performs worse (although honouring safety).

**Summary Of The Paper:**

The paper introduces CAMA, a safety approach for MARL agents based on constraint augmentation of the search space, and especially the reward function under the notion of a safety budget and hazard values.
CAMA is integrated into two MARL paradigms: CTDE and IL.
Experiments are performed to compare CAMA to baselines methods.

**Summary Of The Review:**

The CAMA idea seems good and the results confirm that the safety of the agents is improved.
I have some concerns regarding the degradation of performance in some settings and the actual novelty of the work.

---

> ### Author Response · Authors · 2022-11-18
> **To reviewer CFTd**
>
> **We thank the reviewer CFTd for the valuable suggestions, which will help us improve our paper's quality.**
>
> >**Q1: the reward improvement is not consistent and in some environments CAMA performs worse (although honouring safety).**
> * **Response:** As we wrote in the paper, one of the advantages of CAMA is Plug-and-Play, which also means that CAMA is limited by the original algorithm. For example, the performance of CAMA-HAPPO is better than CAMA-MAPPO in most cases since HAPPO itself have better performance than MACPO in those tasks. This is why we introduced CAMA into both CTDE and IL so that in subsequent applications, the most efficient algorithm in the specific environment can be selected to combine with CAMA in order to satisfy the constraints. It is undeniable that any safety class algorithm will sacrifice some reward, but we can find that CAMA outperforms the benchmark in 4 out of 7 scenarios we tested (Ant 2x4 CTDE & IL, HalfCheetah 3x2 CTDE, Sumo Ant), The remaining three all performed equally or slightly below the benchmark, and such results are achieved under the premise of ensuring safety. We believe that CAMA has potential. Experimenters can achieve more efficient returns in specific tasks by performing finer parameter tuning and replacing the base algorithm.
>
> >**Q2: Details for the implementation of the method are given, although no code release is available.**
> * **Response:** Thank you very much for acknowledging the ease of use of our approach. In the meantime, we will publish the code in the near future. Hopefully, this work will promote more work to focus on the problem of safety constraint satisfaction in MARL.
>
> >**Q3:  I have some concerns regarding the degradation of performance in some settings and the actual novelty of the work.**
> * **Response:** Thanks to the reviewers for their concerns. We would like to re-emphasize the novelty and innovation of CAMA. First of all, this paper claims to extend state augmentation to multi-agent environments, and the technique was not originally designed for safety - implying additional technical attention. For example, it is important to combine safety constraints with each agent in the CTDE and IL settings while maintaining the Markov property. Thanks to our principle derivation and code implementation under the $\boldsymbol{h}^i_{j, t+1}$ variables, CAMA-HAPPO and CAMA-IPPO achieve excellent reward performance while satisfying safety constraints. Furthermore, CAMA is applied to two different game settings, cooperative and competitive, which cannot be covered by previous methods. Therefore, we consider the solved problem setting to be novel, and the results are satisfactory.

---

### Official Review · Reviewer_FCBX · 2022-10-25

**Confidence:** 3
**Correctness:** 3
**Technical Novelty And Significance:** 3
**Empirical Novelty And Significance:** 3
**Recommendation:** 5

**Clarity, Quality, Novelty And Reproducibility:**

The paper is clearly written, with both algorithms and experimental setup quite clearly explained. In terms of contributions,  the main novelty is CAMA, which is a flexible safe RL framework for multi-agent systems. CAMA
can work in both cooperative and competitive multi-agent games. Finally the authors evaluate the CAMA algorithm on
multi-agent control tasks in SMAMujoco and Gym Compete, which shows the algorithms' ability on constraint satisfaction and reward maximisation. Contribution is more on the empirical sense in which a new flexible framework is proposed to solve multi-agent safe RL. Algorithmically the novelty is relatively incremental (in terms of using a standard reward shaping technique to incorporate multiple constraints into the new reward).

**Strength And Weaknesses:**

Strengths:
In this work, the authors proposed the CAMA framework for safety-aware multi-agent RL problems.
CAMA is quite modular and can be used in conjunction with different standard MARL algorithms (to make them safety-aware).
Empirically the authors showed CAMA beats most multi-agent safe RL SOTA in terms of safety and return maximzation.

Weakness:
The idea of safety budget has been well studied by many previous paper, but mainly for single constraint safe RL problem. It seems besides the CAMA framework and experimentations, the main novelty is on using this safety budget for MARL setting, which is quite incremental.


**Summary Of The Paper:**

In this work, the authors introduce the Constraint Augmented Multi-Agent framework — CAMA, which can serve
as a plug-and-play module to the popular MARL algorithms. They propose an approach that represents the safety constraint as discounted safety costs, known as the safety budget and demonstrate in experiments that CAMA converges  quickly to policies with constraint satisfaction and have better return performance when compared with other state-of-the-art safety RL algorithms.

**Summary Of The Review:**

The work is quite neat in terms of addressing the safe RL problem in the MARL setting. The CAMA framework is also quite flexible and can be adapted to different MARL algorithms. Experiments potentially show the superiority of this algorithm in terms of enforcing safety for multi-agent systems as well as effective return maximzation.

However, novelty wise I think the underlying technique is a direct extension of safety budget in standard safe RL, which is quite incremental.

---

> ### Author Response · Authors · 2022-11-18
> **To reviewer FCBX**
>
> **We thank the reviewer FCBX for constructive comments, which will help us improve our paper's quality.**
>
> >**Q1. The idea of safety budget has been well studied by many previous paper, but mainly for single constraint safe RL problem. It seems besides the CAMA framework and experimentations, the main novelty is on using this safety budget for MARL setting, which is quite incremental.**
> * **Response:** Thanks to the reviewer for your concerns. We would like to emphasize the novelty and innovation of CAMA. First of all, the safety problem in MARL itself is a difficult problem and urgent to be solved like we elaborated on in the paper introduction section. Moreover, there is still a lack of safety MARL algorithms with Plug-and-Play properties like CAMA. Second, the paper claims to extend state augmentation to multi-agent environments and that the technique was not originally designed for safety—meaning additional technical attention is required. For example, it is important to incorporate safety constraints with each agent in CTDE and IL settings while maintaining Markov properties. Thanks to our principle derivation and code implementation under the $\boldsymbol{h}^i_{j , t+1}$ variables, CAMA-HAPPO and CAMA-IPPO achieve excellent reward performance while satisfying safety constraints. In addition, CAMA can be applied to two different game settings, cooperative and competitive, which cannot be covered by previous methods. Therefore, we consider the problem-solving setup to be novel, and the results are satisfactory.

---

### Official Review · Reviewer_VkFF · 2022-11-02

**Confidence:** 3
**Clarity, Quality, Novelty And Reproducibility:** The paper is well-written and with de…
**Correctness:** 4
**Technical Novelty And Significance:** 3
**Empirical Novelty And Significance:** 3
**Recommendation:** 6

**Strength And Weaknesses:**

Strengths:
- Having a modular component that can be added to multi-agent reinforcement learning algorithms.
- Having a method that works both with centralized training with decentralized execution and with independent learning.
- The experimental results are extensive. They have been obtained in multiple scenarios and comparing different algorithms.
Weaknesses:
- The contribution is valid but the novelty is limited.
- All the proofs are in an Appendix.

**Summary Of The Paper:**

The paper presents CAMA, a framework for multi-agents that incorporates safety constraints into multi-agent reinforcement learning algorithms. The framework can be added to different multi-agent reinforcement learning algorithms as a plug-and-play method. The safety constraints are represented as the sum of discounted safety costs. Since there is a safety budget, it is easy to compute the hazard value, which is the remaining safety budget. The paper includes empirical results that show the effectiveness of the method.

**Summary Of The Review:**

The paper presents a module that can be added to multi-agent reinforcement learning algorithms to enforce safety constraints. The method is supported by extensive experimental results obtained in simulation.

---

> ### Author Response · Authors · 2022-11-18
> **To reviewer VkFF**
>
> **We thank the reviewer VkFF for the efforts in reviewing our paper, which will help us improve its quality. We have submitted a new revised version and are preparing to make the code public. It is hoped that this work can further advance the application of constraint satisfaction in MARL.**

---

### Decision · Program_Chairs · 2023-01-20

**Decision:**

Reject

**Justification For Why Not Higher Score:**

The biggest problem with this paper is the lack of novelty.  The idea of augmenting the state features with budget features has already been done for single agent RL.  This paper extends this approach to multiple agents.  However, the extension is immediate since there is nothing special to be done about the fact that this is a multiagent system.  The only difference is that the cost function in each constraint may depend on the actions of all agents, but that's it.


**Justification For Why Not Lower Score:**

NA

**Metareview: Summary, Strengths And Weaknesses:**

The paper proposes CAMA, which is a framework for safe multi-agent RL.  In this frameowrk, the state features are augmented with budget features (one budget feature per constraint).  Once a budget runs out, then a negative reward is given to the agent to penalize it since this indicates that the corresponding constraint is violated.  The framework simply changes the MDP formulation, which allows any MDP or RL technique to be used subsequently.

The biggest problem with this paper is the lack of novelty.  The idea of augmenting the state features with budget features has already been done for single agent RL.  This paper extends this approach to multiple agents.  However, the extension is immediate since there is nothing special to be done about the fact that this is a multiagent system.  The only difference is that the cost function in each constraint may depend on the actions of all agents, but that's it.

Strength:
* Generic approach that can leverage all existing multi-agent systems

Weakness:
* Lack of novelty (see above)